# Synthesis of Benzimidazole–Based Analogs as Anti Alzheimer’s Disease Compounds and Their Molecular Docking Studies

**DOI:** 10.3390/molecules25204828

**Published:** 2020-10-20

**Authors:** Bushra Adalat, Fazal Rahim, Muhammad Taha, Foziah J. Alshamrani, El Hassane Anouar, Nizam Uddin, Syed Adnan Ali Shah, Zarshad Ali, Zainul Amiruddin Zakaria

**Affiliations:** 1Department of Chemistry, Hazara University, Mansehra 21300, Pakistan; busraadalat@yahoo.com (B.A.); fazalstar@gmail.com (F.R.); Zarshad11@yahoo.com (Z.A.); 2Department of Clinical Pharmacy, Institute for Research and Medical Consultations (IRMC), Imam Abdulrahman Bin Faisal University, P.O. Box 1982, Dammam 31441, Saudi Arabia; 3Neurology Department, King Fahad Hospital of University, Imam Abdulrahman Bin Faisal University, P.O. Box 1982, Dammam 34211, Saudi Arabia; Fshamrani@iau.edu.sa; 4Department of Chemistry, College of Science and Humanities in Al-Kharj, Prince Sattam Bin Abdulaziz University, Al-Kharj 11942, Saudi Arabia; anouarelhassane@yahoo.fr; 5Department of Chemistry, University of Karachi, Karachi 75270, Pakistan; nizamuddin.abbasi@uok.edu.pk; 6Atta-ur-Rahman Institute for Natural Products Discovery (AuRIns), Universiti Teknologi MARA Cawangan Selangor Kampus Puncak Alam, Bandar Puncak Alam, Selangor 42300, Malaysia; syedadnan@uitm.edu.my; 7Faculty of Pharmacy, Universiti Teknologi MARA Cawangan Selangor Kampus Puncak Alam, Bandar, Puncak Alam, Selangor 42300, Malaysia; 8Department of Biomedical Science, Faculty of Medicine and Health Sciences, Universiti Putra Malaysia, UPM Serdang, Selangor 43400, Malaysia; 9Integrative Pharmacogenomics Institute (iPromise), Universiti Teknologi MARA Cawangan Selangor Kampus Puncak Alam, Bandar Puncak Alam, Selangor 42300, Malaysia

**Keywords:** synthesis, acetylcholinesterase, butyrylcholinesterase, molecular docking, Schiff base, thiosemicarbazide, SAR1

## Abstract

We synthesized **10** analogs of benzimidazole-based thiosemicarbazide **1** (**a**–**j**) and **13** benzimidazole-based Schiff bases **2** (**a**–**m**), and characterized by various spectroscopic techniques and evaluated in vitro for acetylcholinesterase (AchE) and butyrylcholinesterase (BchE) inhibition activities. All the synthesized analogs showed varying degrees of acetylcholinesterase and butyrylcholinesterase inhibitory potentials in comparison to the standard drug (IC_50_ = 0.016 and 4.5 µM. Amongst these analogs **1** (**a**–**j**), compounds 1b, 1c, and 1g having IC_50_ values 1.30, 0.60, and 2.40 µM, respectively, showed good acetylcholinesterase inhibition when compared with the standard. These compounds also showed moderate butyrylcholinesterase inhibition having IC_50_ values of 2.40, 1.50, and 2.40 µM, respectively. The rest of the compounds of this series also showed moderate to weak inhibition. While amongst the second series of analogs **2** (**a**–**m**), compounds 2c, 2e, and 2h having IC_50_ values of 1.50, 0.60, and 0.90 µM, respectively, showed moderate acetylcholinesterase inhibition when compared to donepezil. Structure Aactivity Relation of both synthesized series has been carried out. The binding interactions between the synthesized analogs and the enzymes were identified through molecular docking simulations.

## 1. Introduction

Acetylcholinesterase (AchE) and butyrylcholinesterase (BchE) are enzymes that play an important role in the hydrolysis of acetylcholine into choline and acetic acid [1]. This results in the shortage of acetylcholine in areas such as cortex and hippocampus of brain, which are associated with high psychological functions [2]. These enzymes are responsible for Alzheimer’s disease (AD)—a progressive and an irreversible brain disorder that causes disturbance of the cholinergic system of the brain. This disturbance may cause memory loss, disorientation, cognitive impairment, and difficulty in thinking and problem solving [3,4,5]. In the aging society, AD is the main cause of dementia. These enzymes help in the aggregation of neurotoxic beta amyloid, which causes neuronal cell apoptosis. One of the approaches for treating AD is to target both acetylcholinesterase and butyrylcholinesterase [6,7].

The AchE has two binding sites, a catalytic site that is responsible for acetylcholine hydrolysis and the peripheral site for beta amyloid interaction. The AchE interactions with protein give rise to a complex AchE–Aβ, which causes neurotoxicity. AchE is present in the brain, muscles, and cholinergic neurons while BchE is present in intestine, liver, kidneys, heart, lungs, and serum [8,9]. It plays a main role in the breakdown of ester containing compounds. Normally, AchE is dominant in the brain, while BchE function rises when acetylcholine function gradually decreases in the brain of a patient with Alzheimer’s disease. Therefore, there is an intense need for a drug that can inhibit the activity of both AchE and BchE [10].

For the treatment of Alzheimer’s disease, several drugs have been approved by the FDA such as donepezil, rivastigmine (Figure 1), tacrine, and galantamine [11]. However, hepatotoxicity, insufficient activity, and gastrointestinal disturbance limit the use and applicability of these drugs [12,13,14,15,16]. To overcome the side effect of synthetic ChEs inhibitors, researchers have shown a great interest in the isolation of these compounds as an alternative, and many non-toxic bioactive ChEs inhibitors have been isolated from natural sources [17,18,19]. Amongst these, donepezil and galantamine are selective to AchE, whereas tacrine and rivastigmine inhibit both AchE and BchE. The Benzimidazole nucleus possesses a wide spectrum of activities that ranges from antimicrobial activities to being used against the world most deadly diseases. It has gained importance in medicinal chemistry because of its affinity for different enzymes and protein receptors.

Our research group has been continuously attempting to synthesize new heterocyclic moieties in search of lead therapeutics. Recently, we reported benzimidazole-based oxadiazole as an anti-alpha-glycosidase [20], benzimidazole-bearing bis-Schiff base as alpha glucosidase inhibitor [21,22,23], and 2-mercaptabenzimidazole as alpha amylase inhibitor [24]. Benzimidazole is a potent urease inhibitor [25]. Here, we report two new series of benzimidazole thiosemicarbazides and benzimidazole Schiff bases as novel Alzheimer inhibitors.

## 2. Results and Discussion

### 2.1. Synthesis of Benzimidazole-Based Thiosemicarbazide

We treated 1H-benzimidazole-2-thiol (1) (1-mmol) with methyl (4-bromomethyl) benzoate (**II**) (1-mmol) in ethanol and refluxed it for 3–4 h to give methyl4-(((1H-benzimidazole-2yl)thio) methyl) benzoate (**III**) as an intermediate product. This intermediate was then further treated with hydrazine hydrate in ethanol and refluxed for 5–6 h to give 4-(((1H-benzimidazole-2-yl)thio) methyl) benzohydrazides (**IV**) as a second intermediate product. This intermediate product is then treated with various substituted isothiocyanates which resulted in the final product **1** (**a**–**j**) (Scheme 1, Table 1). The reaction completion was monitored by TLC. The product was filtered and washed with distilled water.

### 2.2. Synthesis of Benzimidazole Schiff Base

In first step, we synthesized benzimidazole analogs by treating 1-methyl-3,4-diaminobenzene (**V**) with ethyl-4-formyl benzoate (**VI**) in dimethyl formide (DMF) along with sodium meta bi-sulfite (Na_2_S_2_O_5_) as a base under reflux condition for 4 to 5 h to yield intermediate (**VII**) as product. The intermediate (**VII**) was treated with benzene hydrate in ethanol and was refluxed for 4–5 h to yield intermediate (**VIII**). In the third step, the intermediate (**VIII**) was treated with various aromatic aldehydes in methanol in the presence of glacial acetic acid to synthesize our final desired products **2** (**a**–**m**) (Scheme 2, Table 2).

### 2.3. Structure Activity Relationship

#### 2.3.1. Acetylcholinesterase Activity of Benzimidazole-Based Thiosemicarbazide

We have synthesized ten analogs **1** (**a**–**j**) of benzimidazole-based thiosemicarbazide and evaluated them for ChE inhibition. All the analogs showed acetylcholinesterase inhibition of varying degrees ranging from 0.60 ± 0.05 µM to 12.90 ± 0.20 µM. All the analogs showed good inhibitory activities. The analogs **1c** and **1g** were the most active in the series. These analogs have chloro groups on the phenyl ring. Analog **1c** having 2 chloro groups at position 2 and 3 of phenyl ring has an IC_50_ value of 0.60 ± 0.05 µM, while in analog **1g**, the chloro groups are at position 3 and 4 with the IC_50_ value of 0.80 ± 0.50 µM. These results showed that enzyme inhibition activity mainly depends on the substitution pattern of chloro group on the phenyl ring. Similarly, analog **1b** having floro group at position 4 on phenyl ring also showed a good inhibition with an IC_50_ value of 1.30 ± 0.01 µM, while analog **1a** having bromo group at position 2 of phenyl ring has an IC_50_ value of 11.30 µM. This shows that inhibition activity not only depends on the position of substituent but also on the number of substituents attached to the phenyl ring. The analog **1h** having one methyl group at position 4 has an IC_50_ value of 3.50 ± 0.01 µM, while analog **1i** having methyl group at 3 position has an IC_50_ value of 7.30 ± 0.10 µM. These analogs also showed good inhibition. The difference in their inhibition values may be due to the difference in the position of methyl group on the phenyl ring. The analog **1d** having 2 methyl groups—one at position 2 and the other at 6—has an IC_50_ value of 5.60 µM. The analog **1f** having nitro group at position 3 has an IC_50_ value of 6.20 µM and analog 5 having nitro group at position 4 has an IC_50_ value of 8.50 µM. The analog **1j** having phenyl ring has an IC_50_ value of 12.96 µM.

The analogs **1**(**a**–**j**) showed varying degrees of butyrylcholinesterase inhibition ranging from 1.50 ± 0.10 to 29.10 ± 0.30 µM when compared to the standard drug donepezil having an IC_50_ value of 4.5 ± 0.11 µM. The most active analogs among the series were **1b**, **1c**, **1e**, **1f**, and **1g** that showed excellent butyrylcholinesterase inhibition with IC_50_ values of 2.40 ± 0.10, 1.50 ± 0.10, 10.30 ± 0.20, 12.90 ± 0.20 µM, and 2.40 ± 0.10 µM, respectively. The analogs **1b, 1c**, and **1g** were the most active. These analogs have halogen atoms on different positions of the phenyl ring. This shows that the presence and position of halogen atoms on phenyl ring plays a major role in inhibition. Analog **1a** showed moderate inhibition with the IC_50_ value of 22.60 ± 0.50 µM. This analog has a bromine atom at position 2 on the phenyl ring. Analogs **1e** and **1f** have a nitro group on the phenyl ring. The slight difference in the inhibition value may be due to their different positions on the phenyl ring. Similarly, analogs **1h** and **1i** having a methyl group on the phenyl ring have an IC_50_ value of 12.80 ± 0.20 µM and 13.40 ± 0.20 µM, respectively. Both these analogs have electron donating groups at different positions on the phenyl ring. The analog **1d** showed a poor inhibition with an IC_50_ value of 29.10 ± 0.30 µM. This analog has 2 methyl groups at position 2 and 6, respectively. Another least active analog was analog **1j** having a phenyl ring with an IC_50_ value of 24.40 ± 0.30 µM.

#### 2.3.2. Acetylcholinesterase Inhibition Activity of Benzimidazole Schiff Bases

The synthesized benzimidazole Schiff base analogs were examined for AchE and BchE. All the compounds exhibited good to moderate activity when compared with the standard drug donepezil having an IC_50_ value of AchE 0.016 ± 0.12 µM and BchE 4.5 ± 0.11 µM. Analog **2e** was found to be the most active analog among the series, showing a good inhibition of both AchE and BchE with the IC_50_ values of 0.60 ± 0.05 µM and 2.20 ± 0.10 µM, respectively. This analog has two chlorine atomsone at position 3 and the other at 4of the phenyl ring. Similarly, analog **2c** also exhibited significant inhibition of AchE and BchE with an IC_50_ value of 1.50 ± 0.10 µM and 4.10 ± 0.10 µM, respectively. This analog has two chlorine atoms—one at position 3 and the other at 4—and a hydroxyl group at position 2. The ChE inhibition data in Table 1 show that the substitution of chlorine atoms on the phenyl ring plays a significant role in inhibition. The next active analog among the series is analog **2h** having a dimethyl amino group at position 4 on the phenyl ring. This analog also showed a significant inhibition of both AchE IC_50_ = 0.90 ± 0.50 µM and BchE IC_50_ = 2.20 ± 0.10 µM.

Similarly, the inhibitory potential of the compounds having hydroxyl group and methoxy group on phenyl ring was also good, e.g., analog **2l** has a hydroxyl group at position 4 on the phenyl ring and a methoxy group at positions 3 and 5, and exhibited inhibition for AchE and BchE with IC_50_ values of 2.30 ± 0.10 µM and5.15 ± 0.10 µM, respectively. Analog **2f** having a hydroxyl group at position 3 and a methoxy group at position 4 showed a selective inhibition toward AchE with an IC_50_ value of 4.20 ± 0.1 µM, and for BchE, the IC_50_ = 7.60 ± 0.10 µM. Analog **2d** having a hydroxyl group on the naphthalene ring also showed a good inhibition toward AchE, IC_50_ = 3.10 ± 0.10 µM, and BchE, IC_50_ = 6.20 ± 0.10 µM. This shows that the hydroxyl group plays an important role in inhibition. Similarly, analog **2a** and **2b** having a nitro group on the phenyl ring were more selective in inhibition toward AchE with IC_50_ values 3.40 ± 0.10 µM and 8.50 ± 0.20 µM, while for BchE they showed a weak inhibition with IC_50_ values of 7.50 ± 0.30 µM and12.10 ± 4.0 µM, respectively. The rest of the compounds of the series showed weak inhibitory activities.

### 2.4. Molecular Docking Studies

The inhibition concentration values of benzimidazole thiosemicarbazide derivatives **1 (a**–**j**) and benzimidazole Schiff bases **2** (**a**–**m**) as effective acetylcholinesterase inhibitors are displayed in Table 3. The acetylcholinesterase inhibition by the synthesized derivatives may be influenced by the type, number, and positions of the substituted functional groups of their basic skeletons (Table 3). To rationalize the observed acetylcholinesterase inhibition by the synthesized derivatives, molecular docking has been carried out to determine the binding modes between the synthesized derivatives **1** (**a**–**j**) and **2** (**a**–**m**) from one side and the active residues of the acetylcholinesterase from another side. Table 3 gathers the calculated free binding energies of the stable complexes’ ligand–acetylcholinesterase, the number of established intermolecular hydrogen bonding between the synthesized compounds and active site residues of acetylcholinesterase, and the closest residues to the docked compounds and their IC_50_ values.

All the complexes formed between the synthesized derivatives **1 (a**–**j**) and **2** (**a**–**m**) and the active residues of acetylcholinesterase displayed negative binding energies, which demonstrates that acetylcholinesterase inhibition by synthesized derivatives **1 (a**–**j**) and **2** (**a**–**m**) is a thermodynamic favorable process (Table 3). The docking of the best active benzimidazole thiosemicarbazides (**1c**, **1b** and **1g**), benzimidazole Schiff bases (**2c**, **2h** and **2e**), and donepezil into the active binding site of acetylcholinesterase are displayed in Figure 2.

For benzimidazole thiosemicarbazide derivatives, binding energies of the stable complexes vary with a maximal variation of 1.56 kcal mol^−1^. In accordance with the MIC values, the best active compound **1c** with the IC_50_ value of 0.60 µM has the lowest binding energy of −13 kcal/mol. These variations are low enough to be considered as a strong descriptor in rationalizing the observed acetylcholinesterase inhibition. However, the number of hydrogen bonding, its distances, and the intermolecular interactions between the substitute groups of **1 (a**–**j**) and the active residues may help in understanding the observed acetylcholinesterase inhibition. For instance, the compounds **1e** and **1f** differ by the position of the substituted nitro group at the aromatic ring, where in the former the nitro group is substituted at *para* position while in the latter it is substituted at *meta* position (Scheme 1). Experimentally, **1e** shows a higher activity than **1f**. Their corresponding complexes formed with acetylcholinesterase display similar binding energies with a variation less than 0.01 kcal/mol^−1^. The higher acetylcholinesterase inhibition of **1e** compared with **1f** may refer to the number of hydrogen bonding that was established by the former with the active residues of acetylcholinesterase compared with the latter (Figure 3). As can be seen in Figure 3 and Table 3, in **1e**, the nitro group at *para* position forms three hydrogen bonds with amino acids GLY B:122, ALA B:204, and GLY B:121 of acetylcholinesterase with distances 2.79, 3.00, and 3.23 Å, respectively. However, in **1f**, the nitro group at *meta* position forms one hydrogen bonding with TYR B:124 with a distance of 2.85 Å.

For benzimidazole Schiff bases, binding energies of the stable complexes vary with variations in the range 0–8.34 kcal/mol^−1^ with respect to the stable one. Similar to the benzimidazole thiosemicarbazide derivatives, the potency of benzimidazole Schiff bases to inhibit acetylcholinesterase is strongly related to the substituted functional groups, which may increase the stability of benzimidazole Schiff bases–acetylcholinesterase complexes. For instance, the compounds **2a** and **2b** differ by the position of the substituted nitro group at the aromatic, where in the former the nitro group is substituted at *ortho* position while in the latter it is substituted at *meta* position (Scheme 2). Experimentally, **2a** shows a higher activity than **2b**. The higher acetylcholinesterase inhibition of **2a** compared with **2b** may refer to the stability of the complex formed by the former with acetylcholinesterase, which displays a binding energy variation of 1.35 kcal/mol^−1^ compared to the complex formed with **2b**. The stability of 2a–acetylcholinesterase may refer to the number of hydrogen bonds **2a** established with the active residues of acetylcholinesterase compared with **2b** (Figure 4). As can be seen in Figure 4 and Table 3, in 2a–acetylcholinesterase, five hydrogen bonds are formed between **2a** and amino acids ASP B:74, TYR B:72, SER B:203, GLY B:122 and SER B:125, while in **2b**–acetylcholinesterase, three hydrogen bonds are formed between **2b** and amino acids ALA B:204, GLY B:121 and GLY B:122.

## 3. Materials and Methods

### 3.1. General Procedure for the Synthesis of Benzimidazole-Based Thiosemicarbazide 1*(**a**–**j**)*

We treated 1H-benzimidazole-2-thiol (**1**) (1-mmol) with methyl (4-bromomethyl) benzoate (**II**) (1-mmol) in ethanol and refluxed it for 3–4 h to give methyl4-(((1H-benzimidazole-2yl)thio) methyl) benzoate (**III**) as an intermediate product. This intermediate was then further treated with hydrazine hydrate in ethanol and refluxed for 5–6 h to give 4-(((1H-benzimidazole-2-yl)thio) methyl) benzohydrazides (**IV**) as a second intermediate product. This intermediate product is then treated with various substituted isothiocyanate which gave the final product **1**(**a**–**j**). The reaction completion was monitored by TLC. The product was filtered and washed with distilled water.

*2-(4-(((1H-benzo[d]imidazol-2-yl)thio)methyl)benzoyl)-N-(2-bromophenyl)hydrazine-1-carbothioamide* (**1a**): Yield (88%); m.p.: 314–315 °C. IR ν cm^−1^ (KBr disk): 3460 cm^−1^ (NH stretch), 3163 cm^−1^ (Ar.C-H stretch), 1641 cm^−1^ (C = O), 1582 cm^−1^ (C = N), 1552 cm^−1^ (C-N), 550 (C-Br); ^1^HNMR (500 MHz, DMSO-d_6_): δ 12.58 (S,1H, NH), 10.5 (S,1H, NH), 9.8 (S,1H, NH), 9.5 (S,1H, NH), 7.89 (d, *J* = 5.8 Hz, 2H, Ar), 7.62 (d, *J* = 6.2 Hz, 1H, Ar), 7.57(m, 4H, Ar), 7.37 (t, *J* = 6.3 Hz, 2H, Ar), 7.19 (t, *J* = 6.2 Hz, 1H, Ar), 7.12(t, *J* = 6.3 Hz, 2H, Ar), 4.65 (S, 2H, S-CH_2_). ^13^CNMR (125 MHz, DMSO-*d*_6_): δ 182.3, 164.8, 150.2, 144.1, 140.3, 139.2, 139.3, 137.3, 132.7, 132.0, 130.2, 128.6, 127.3 (2C), 127.6 (2C), 122.8 (2C), 121.9, 116.4, 116.2, 34.5.

*2-(4-(((1H-benzo[d]imidazol-2-yl)thio)methyl)benzoyl)-N-(4-fluorophenyl)hydrazine-1-carbothioamide* (**1b**): Yield (84%); m.p.: 319–320 °C. IR ν cm^−1^ (KBr disk): 3490 cm^−1^ (NH stretch), 3185 cm^−1^ (Ar.C-H stretch), 1645 cm^−1^ (C = O), 1582 cm^−1^ (C = N), 1552 cm^−1^ (C-N), 1050 (C-F); ^1^HNMR (500 MHz, DMSO-*d*_6_): 12.58 (S,1H, NH), 10.49 (S,1H, NH), 9.73 (S,2H, NH), 7.88 (d, *J* = 6.4 Hz, 2H, Ar), 7.57 (m, 4H, Ar), 7.36 (t, *J* = 5.9 Hz, 3H,Ar), 7.12 (t, *J* = 6.3 Hz, 3H,Ar), 4.65 (S, 2H,S-CH_2_). ^13^CNMR (125 MHz, DMSO-*d*_6_): δ 182.6, 165.1, 163.2, 150.1, 143.9, 140.8, 138.4, 138.6, 134.6, 130.4, 130.3, 127.4(2C), 127.1, 127.0, 123.3, 123.1, 116.3, 116.2, 115.9, 115.7, 34.6.

*2-(4-(((1H-benzo[d]imidazol-2-yl)thio)methyl)benzoyl)-N-(2,3-dichlorophenyl) hydrazine-1-carbothioamide* (**1c**): Yield (79%); m.p.: 321–322 °C. IR ν cm^−1^ (KBr disk): 3396 cm^−1^ (NH stretch), 3159 cm^−1^ (Ar.C-H stretch), 1649 cm^−1^ (C = O), 1571 cm^−1^ (C = N), 1550 cm^−1^ (C-N), 745 (C-Cl); ^1^HNMR (500 MHz, DMSO-*d*_6_): 12.57(S, 1H, NH), 10.58 (S, 1H, NH), 9.92 (S, 1H, NH), 9.75 (S, 1H, NH) 7.88 (d, *J* = 6.2 Hz, 2H, Ar), 7.57 (t, *J* = 6.9 Hz, 4H, ArH), 7.35 (t, *J* = 5.95 Hz, 3H, Ar), 7.12 (t, *J* = 6.2 Hz, 2H, ArH), 4.61(S, 2H, S-CH_2_). ^13^CNMR (125 MHz, DMSO-*d*_6_): δ 183.2, 165.2, 150.1, 145.1, 140.9, 139.2, 139.1, 138.2, 137.4, 132.9, 132.2, 129.8, 128.0, 127.5, 127.3, 127.3, 127.2, 123.4, 123.2, 116.1, 116.3, 34.5.

*2-(4-(((1H-benzo[d]imidazol-2-yl)thio)methyl)benzoyl)-N-(2,6-dimethylphenyl) hydrazine-1-carbothioamide* (**1d**): Yield (81%); m.p.: 311–312 °C. IR ν cm^−1^ (KBr disk): 3396 cm^−1^ (NH stretch), 3165 cm^−1^ (Ar.C-H stretch), 1650 cm^−1^ (C = O), 1576 cm^−1^ (C = N), 1548 cm^−1^ (C-N); ^1^HNMR (500 MHz, DMSO-*d*_6_): 12.58 (S, 1H, NH), 10.49 (S, 1H, NH), 9.54 (S, 1H, NH), 9.35 (S, 1H, NH), 7.8 (d, *J* = 6.6 Hz, 2H, Ar), 7.54 (m, 3H, Ar), 7.35 (d, *J* = 6.1 Hz, 2H, Ar), 7.23(m, 3H, Ar), 7.1 (m, 1H, Ar), 4.61(S, 2H, S-CH_2_). ^13^CNMR (125 MHz, DMSO-*d*_6_): δ 182.5, 165.2, 149.9, 143.5, 141.7, 138.8, 138.6, 135.8, 134.9, 134.7, 128.1, 128.3, 127.6(2C), 127.4, 127.3, 127.0, 122.8, 122.7, 115.6, 115.5, 34.7, 18.4(2C).

*2-(4-(((1H-benzo[d]imidazol-2-yl)thio)methyl)benzoyl)-N-(4-nitrophenyl)hydrazine-1-carbothioamide* (**1e**): Yield (78%); m.p.: 319–320 °C. IR ν cm^−1^ (KBr disk): 3426 cm^−1^ (NH stretch), 3195 cm^−1^ (Ar.C-H stretch), 1652 cm^−1^ (C = O), 1579 cm^−1^ (C = N), 1546 cm^−1^ (C-N), 1530 (NO_2_), 1360 (NO_2_); ^1^HNMR (500 MHz, DMSO-*d*_6_): 12.6 (S, 1H, NH), 10.58 (S, 1H, NH), 10.12 (S, 1H, NH), 10.07 (S, 1H, NH), 8.20 (d, *J* = 6.9 Hz, 2H, Ar), 7.8 (d, *J* = 5.61 Hz, 4H, Ar), 7.5 (m, 3H, Ar), 7.1 (m, 3H, Ar), 4.63(S, 2H, S-CH_2_). ^13^CNMR (125 MHz, DMSO-*d*_6_): δ 182.2, 165.4, 149.5, 145.6, 144.9, 144.1, 140.6, 139.3, 139.2, 127.8, 127.7, 127.4, 127.3, 124.8, 124.7, 124.1, 124.0, 123.9, 123.7, 115.8, 115.6, 34.8.

*2-(4-(((1H-benzo[d]imidazol-2-yl)thio)methyl)benzoyl)-N-(3-nitrophenyl)hydrazine-1-carbothioamide* (**1f**): Yield (83%); m.p.: 3316–317 °C. IR ν cm^−1^ (KBr disk): 3412 cm^−1^ (NH stretch), 3175 cm^−1^ (Ar.C-H stretch), 1647 cm^−1^ (C = O), 1575 cm^−1^ (C = N), 1542 cm^−1^ (C-N), 1533 (NO_2_), 1355 (NO_2_); ^1^HNMR (500 MHz, DMSO-*d*_6_): 10.60 (S, 1H, NH), 10.06 (S, 1H, NH), 8.4 (S, 1H, NH), 8.0 (S, 1H, NH), 7.9 (d, *J* = 6.3 Hz, 2H, Ar), 7.8 (d, *J* = 6.9 Hz, 3H, Ar), 7.68 (m, 4H, Ar), 7.13 (m, 3H, Ar), 4.64 (S, 2H, S-CH_2_). ^13^CNMR (125 MHz, DMSO-*d*_6_): δ 182.7, 165.6, 149.2, 147.9, 142.1, 140.4, 139.1, 139.2, 138.3, 131.1, 129.1, 128.8, 128.7, 128.4, 128.0, 122.4, 122.2, 119.6, 119.4, 109.8, 109.8, 34.7.

*2-(4-(((1H-benzo[d]imidazol-2-yl)thio)methyl)benzoyl)-N-(3,4-dichlorophenyl)hydrazine-1-carbothioamide* (**1g**) Yield: (78%); m.p.: 312–313 °C. IR ν cm^−1^ (KBr disk): 3405 cm^−1^ (NH stretch), 3189 cm^−1^ (Ar.C-H stretch), 1652 cm^−1^ (C = O), 1576 cm^−1^ (C = N), 1554 cm^−1^ (C-N), 780 (C-Cl); ^1^HNMR (500 MHz, DMSO-*d*_6_): 12.59 (S, 1H, NH), 10.54 (S, 1H, NH), 9.97 (S, 1H, NH), 9.85 (S, 1H, NH), 7.8 (d, *J* = 6.6 Hz, 2H, Ar), 7.7 (d, *J* = 6.9 Hz, 1H, Ar), 7.6 (m, 4H, Ar), 7.5 (d, *J* = 6.5 Hz, 2H, Ar) 7.1 (m, 2H, Ar), 4.62 (S, 2H, S-CH_2_). ^13^CNMR (125 MHz, DMSO-*d*_6_): δ 182.6, 165.5, 149.3, 142.1, 139.3, 137.8, 137.7, 135.6, 134.9, 131.2, 129.7, 129.6, 128.6, 128.6, 128.1, 128.0, 121.5, 121.6, 121.2, 116.3, 116.1, 34.6.

*2-(4-(((1H-benzo[d]imidazol-2-yl)thio)methyl)benzoyl)-N-(p-tolyl)hydrazine-1-carbothioamide* (**1h**): Yield (80%); m.p.: 323–324 °C. IR ν cm^−1^ (KBr disk): 3360 cm^−1^ (NH stretch), 3180 cm^−1^ (Ar.C-H stretch), 2960 (C-CH_3_), 1645 cm^−1^ (C = O), 1574 cm^−1^ (C = N), 1555 cm^−1^ (C-N); ^1^HNMR (500 MHz, DMSO-*d*_6_): 12.59 (S, 1H, NH), 10.46 (S, 1H, NH), 9.68 (S, 1H, NH), 7.8 (d, *J* = 6.1 Hz, 2H, Ar), 7.5 (d, *J* = 6.3 Hz, 4H, Ar), 7.27 (d, *J* = 6.2 Hz, 2H, Ar), 7.11(d, *J* = 5.9 Hz, 4H, Ar), 4.65 (S, 2H, S-CH_2_). ^13^CNMR (125 MHz, DMSO-*d*_6_): δ 181.9, 165.6, 149.3, 141.9, 139.5, 137.2, 137.5, 136.6, 134.1, 129.7, 129.4, 128.5(2C), 128.3, 128.2, 125.9, 125.8, 121.2, 121.3, 117.3, 117.0, 34.7, 25.0.

*2.-(4-(((1H-benzo[d]imidazol-2-yl)thio)methyl)benzoyl)-N-(m-tolyl)hydrazine-1-carbothioamide* (**1i**) Yield: (76%); m.p.: 313–314 °C. IR ν cm^−1^ (KBr disk): 3410 cm^−1^ (NH stretch), 3160 cm^−1^ (Ar.C-H stretch), 2980 (C-CH_3_), 1648 cm^−1^ (C = O), 1571 cm^−1^ (C = N), 1552 cm^−1^ (C-N); ^1^HNMR (500 MHz, DMSO-*d*_6_): 11.0 (S, 1H, NH), 10.47 (S, 1H, NH), 9.7 (S, 1H, NH), 9.6 (S, 1H, NH), 7.8 (d, *J* = 6.4 Hz, 2H, Ar), 7.5 (d, *J* = 6.6 Hz, 4H, Ar), 7.19 (m, 4H, Ar) 6.9 (d, *J* = 5.9 Hz, 2H, Ar), 4.6 (S, 2H, S-CH_2_). ^13^CNMR (125 MHz, DMSO-*d*_6_): δ 182.2, 165.5, 149.6, 141.7, 139.0, 137.9, 137.7, 137.5, 136.8, 129.6, 128.7, 128.6, 128.1, 128.0, 125.6, 125.5, 123.3, 123.0, 122.8, 117.8, 117.6, 34.7, 23.9.

*2-(4-(((1H-benzo[d]imidazol-2-yl)thio)methyl)benzoyl)-N-phenylhydrazine-1-carbothioamide* (**1j**) Yield: (77%); m.p.: 309–310 °C. IR ν cm^−1^ (KBr disk): 3405 cm^−1^ (NH stretch), 3186 cm^−1^ (Ar.C-H stretch), 1651 cm^−1^ (C = O), 1572 cm^−1^ (C = N), 1550 cm^−1^ (C-N); ^1^HNMR (500 MHz, DMSO-d_6_): 12.59 (S, 1H, NH), 10.4 (S, 1H, NH), 9.75 (S, 1H, NH), 9.69 (S, 1H, NH), 7.8 (d, *J* = 6.5 Hz, 2H, Ar), 7.5 (d, *J* = 6.7 Hz, 4H, Ar), 7.3 (t, *J* = 6.1 Hz, 2H, Ar), 7.1(m, 5H, Ar), 4.6 (S, 2H, S-CH_2_). ^13^CNMR (125 MHz, DMSO-*d*_6_): δ 10:182.7, 165.6, 149.2, 143.5, 140.2, 138.6, 138.3, 136.7, 128.8, 128.9, 128.6, 128.7, 128.1, 128.2, 127.3, 124.5(2C), 122.9, 122.8, 116.8, 116.5, 34.7.

### 3.2. General Procedure for the Synthesis of Benzimidazole-Based Schiff Bases 2*(**a**–**m**)*

In first step, we synthesized benzimidazole analogs by treating 1-methyl-3,4-diaminobenzene (**V**) with ethyl-4-formyl benzoate (**VI**) in dimethyl formide (DMF) along with sodium meta bi-sulfite (Na_2_S_2_O_5_) as a base under reflux condition for 4 to 5 h to yield intermediate (**VII**) as product. The intermediate (**VII**) was treated with benzene hydrate in ethanol and was refluxed for 4–5 h to yield intermediate (**VIII**). In the third step, the intermediate (**VIII**) was treated with various aromatic aldehydes in methanol in the presence of glacial acetic acid to synthesize our final desired products **2** (**a**–**m**).

*4-(5-methoxy-1H-benzo[d]imidazol-2-yl)-N′-(2-nitrobenzylidene)benzohydrazide* (**2a**): Yield (81%); m.p.: IR ν cm^−1^ (KBr disk): 3365 cm^−1^ (NH stretch), 3174 cm^−1^ (Ar.C-H stretch), 1653 cm^−1^ (C = O), 1572 cm^−1^ (C = N), 1551cm^−1^ (C-N), 1525 (NO_2_), 1345 (NO_2_) 1160 (C-O-C); ^1^HNMR (500 MHz, DMSO-*d*_6_): 12.3 (S,1H, NH), 9.6 (S,1H, NH), 8.3 (d, *J =* 6 Hz, 2H, Aromatic H), 8.1 (d, *J* = 6.6 Hz, 2H,Aromatic H), 8.0 (m, 3H, Aromatic H), 7.7 (d, *J* = 6.5 Hz, 2H, Aromatic H), 7.6 (S,1H, Aromatic H), 7.4 (d, *J* = 6.7 Hz,1H, Aromatic H), 3.5 (S, 3H, OCH_3_). ^13^CNMR (125 MHz, DMSO-*d*_6_): δ 164.4, 157.6, 153.3, 147.7, 142.5, 140.2, 138.6, 135.1, 134.5, 133.7, 132.4, 130.8, 130.7, 130.2, 129.5, 128.1, 128.0, 124.8, 117.8, 112.4, 10I.3, 57.3.

*4-(5-methoxy-1H-benzo[d]imidazol-2-yl)-N′-(3-nitrobenzylidene)benzohydrazide* (**2b**): Yield (84%); m.p.: 311–312 °C. IR ν cm^−1^ (KBr disk): 3360 cm^−1^ (NH stretch), 3181 cm^−1^ (Ar.C-H stretch), 1654 cm^−1^ (C = O), 1578 cm^−1^ (C = N), 1557 cm^−1^ (C-N), 1528 (NO_2_), 1348 (NO_2_) 1168 (C-O-C); ^1^HNMR (500 MHz, DMSO-*d*_6_): 11.6 (S,1H, NH), 9.9 (S,1H, N = CH), 8.8(S,1H, Aromatic H), 8.5(S,1H, Aromatic H), 8.3(d, *J* = 6.8 Hz,1H, Aromatic H), 8.1(m, 3H, Aromatic H), 7.7 (d, *J* = 6 Hz, 2H Aromatic H), 7.6 (m, 2H Aromatic), 7.5(d, *J* = 6.3 Hz,1H Aromatic H), 3.5(S, 3H, OCH_3_). ^13^CNMR (125 MHz, DMSO-d_6_): δ 164.8, 157.9, 154.1, 147.5, 145.8, 140.1, 138.9, 135.0, 134.8, 132.9, 131.9, 130.6, 130.5, 129.8, 128.3, 128.3, 126.0, 120.8, 116.9, 110.8, 102.5, 56.7.

*N′-(3,4-dichloro-2-hydroxybenzylidene)-4-(5-methoxy-1H-benzo[d]imidazol-2-yl) benzohydrazide* (**2c**): Yield (85%); m.p.: 319–320 °C. IR ν cm^−1^ (KBr disk): 3570 cm^−1^ (OH stretch), 3355 cm^−1^ (NH stretch), 3160 cm^−1^ (Ar.C-H stretch), 1643 cm^−1^ (C = O), 1576 cm^−1^ (C = N), 1554 cm^−1^ (C-N), 1180 (C-O), 768 (C-Cl); ^1^HNMR (500 MHz, DMSO-*d*_6_): 11.7 (S, 1H, NH), 8.7(S,1H, NH), 8.6 (S, 1H, OH), 8.3(S,1H,N = CH), 8.2(d, *J* = 6 Hz, 2H, Aromatic H), 8.1(d, *J* = 6.7 Hz, 2H, Aromatic H), 8.0(d, *J* = 6.9 Hz, 2H, Aromatic) 7.3(m,1H, Aromatic H), 3.8 (S, 3H, OCH_3_). ^13^CNMR (125 MHz, DMSO-*d*_6_): δ 164.5, 162.3, 158.1, 153.2, 147.8, 140.3, 139.1, 138.6, 134.9, 133.3, 130.6, 130.6, 129.9, 127.9, 127.8, 125.2, 123.6, 119.2, 117.4, 111.8, 101.4, 56.8.

*N′-((3-hydroxynaphthalen-2-yl)methylene)-4-(5-methoxy-1H-benzo[d]imidazol-2-yl)benzohydrazide* (**2d**) Yield: (77%); m.p.: 304–305°C. IR ν cm^−1^ (KBr disk): 3540 cm^−1^ (OH stretch), 3320 cm^−1^ (NH stretch), 3174 cm^−1^ (Ar.C-H stretch), 1649 cm^−1^ (C = O), 1579 cm^−1^ (C = N), 1558 cm^−1^ (C-N), 1140 (C-O); ^1^HNMR (500 MHz, DMSO-*d*_6_): 12.7(S, 1H, NH), 9.9(br s, 1H, 0H), 9.8(S, 1H, NH), 9.5(S, 1H, N = CH), 8.4(d, *J* = 6.4 Hz, 2H, Aromatic H) 8.3(d, *J* = 5.8 Hz, 1H, Aromatic H), 7.9(m, 4H, Aromatic H), 7.8(d, *J* = 6.6 Hz, 2H, Aromatic H), 7.5(m, 2H, Aromatic H), 7.3(d, *J* = 6.4 Hz, 1H), 7.1(d, *J* = 7.2 Hz, 1H, Aromatic H), 3.9(S, 3H, OCH_3_). ^13^CNMR (125 MHz, DMSO-*d*_6_): δ 164.4, 158.2, 157.5, 153.1, 146.8, 140.3, 139.8, 138.6, 134.1, 132.9, 129.8, 129.7, 128.2, 127.4, 127.5, 126.3, 125.8, 124.9, 123.5, 117.3, 116.2, 111.8, 111.3, 105.9, 100.8, 56.3.

*N’-(3,4-dichlorobenzylidene)-4-(5-methoxy-1H-benzo[d]imidazol-2-yl)benzohydrazide* (**2e**): Yield (72%); m.p.: 306–307 °C. IR ν cm^−1^ (KBr disk): 3415 cm^−1^ (NH stretch), 3176 cm^−1^ (Ar.C-H stretch), 1646 cm^−1^ (C = O), 1579 cm^−1^ (C = N), 1558 cm^−1^ (C-N), 1143 (C-O), 790 (C-Cl); ^1^HNMR (500 MHz, DMSO-*d*_6_): 12.1(S, 1H, NH), 9.5(S, 1H, NH), 8.6(S, 1H, N = CH), 8.4(br s, 1H Aromatic H), 8.3 (d, *J* = 6 Hz, 2H, Aromatic H), 8.1(m, 3H, Aromatic H), 7.7(d, *J* = 6.6 Hz, 2H, Aromatic H), 7.5(S, 1H, Aromatic H), 7.4(d, *J* = 6.4 Hz,1H, Aromatic H), 3.7(S, 3H, OCH_3_). ^13^CNMR (125 MHz, DMSO-*d*_6_): δ 163.9, 156.8, 153.1, 140.4, 139.6, 137.9, 135.2, 134.2, 133.8, 133.4, 132.9, 130.6, 130.1, 129.8, 129.9, 128.1, 127.6, 127.6, 115.8, 110.5, 100.6, 56.9.

*N′-(3-hydroxy-4-methoxybenzylidene)-4-(5-methoxy-1H-benzo[d]imidazol-2-yl)benzohydrazide* (**2f**); Yield (87%); m.p.: 312–313°C. IR ν cm^−1^ (KBr disk): 3563 cm^−1^ (OH stretch), 3326 cm^−1^ (NH stretch), 3171 cm^−1^ (Ar.C-H stretch), 1653 cm^−1^ (C = O), 1567 cm^−1^ (C = N), 1551 cm^−1^ (C-N), 1172 (C-O); ^1^HNMR (500 MHz, DMSO-*d*_6_): 11.9(S, 1H, NH), 9.9(S, 1H, NH), 8.3(br s, 1H, OH), 8.1(S, 1H, N = CH), 7.7(d, *J* = 6.7 Hz, 2H, Aromatic H), 7.6(m, 2H, Aromatic H), 7.4(S,1H, Aromatic H), 7.2(m, 2H, Aromatic H), 6.8(d, *J* = 6.8 Hz, 1H, Aromatic H), 6.6(m, 2H, Aromatic H), 3.8(S,6H, OCH_3_). ^13^CNMR (125 MHz, DMSO-*d*_6_): δ 164.7, 156.2, 153.0, 152.6, 148.2, 146.9, 140.5, 138.6, 134.2, 133.0, 130.9, 130.1, 130.0, 127.7, 127.6, 123.0, 116.2, 115.3, 112.8, 111.2, 101.4, 57.4, 56.2.

*N′-(anthracen-9-ylmethylene)-4-(5-methoxy-1H-benzo[d]imidazol-2-yl)benzohydrazide* (**2g**): Yield (83%); m.p.: 307–308 °C. ^1^HNMR (500 MHz, DMSO-*d*_6_): 11.7(s,1H, NH), 10.7(s,1H,NH), 8.2(s, 1H, N = CH), 8.1(d, *J* = 7.3 Hz, 2H, Aromatic H), 7.8(m, 2H, Aromatic H), 7.6(d, *J* = 7 Hz, 2H, Aromatic H), 7.4(d, *J* = 6.8 Hz, 1H, Aromatic H), 7.3(m, 4H, Aromatic H), 7.0(m, 3H, Aromatic H), 6.9(s,1H, Aromatic H), 6.8(d, *J* = 6 Hz, 1H, Aromatic H), 3.7(s, 3H, OCH_3_). ^13^CNMR (125 MHz, DMSO-*d*_6_): δ 164.2, 157.1, 152.9, 143.0, 140.2, 137.9, 134.2, 132.8, 132.1, 132.0, 130.1, 130.0, 128.9, 128.9, 128.8, 128.2, 128.2, 128.1, 128.1, 127.6, 127.5, 125.8, 125.8, 125.5, 125.5, 124.1, 116.8, 112.1, 100.9, 56.3.

*N′-(4-(dimethylamino)benzylidene)-4-(5-methoxy-1H-benzo[d]imidazol-2-yl)benzohydrazide* (**2h**): Yield (77%); m.p.: 317–318 °C. IR ν cm^−1^ (KBr disk): 3410 cm^−1^ (NH stretch), 3171 cm^−1^ (Ar.C-H stretch), 1658 cm^−1^ (C = O), 1572 cm^−1^ (C = N), 1554 cm^−1^ (C-N), 1147 (C-O); ^1^HNMR (500 MHz, DMSO-*d*_6_): 12.8(s, 1H, NH), 11.6(s, 1H, NH), 8.5(s, 1H, N = CH), 8.0(d, *J* = 6 Hz, 2H, Aromatic H), 7.5(d, *J* = 6.5 Hz, 1H, Aromatic H), 7.4(m, 4H, Aromatic H), 7.3(s, 1H, Aromatic H), 6.8(d, *J* = 6.5 Hz, 2H, Aromatic H), 6.7(d, *J* = 6.6 Hz, 1H, Aromatic), 3.0(s, 3H, OCH_3_), 2.9(6H, N-(CH_3_)_2_). ^13^CNMR (125 MHz, DMSO-*d*_6_): δ 164.3, 157.0, 153.7, 152.8, 147.1, 140.1, 137.8, 134.6, 132.6, 130.2, 130.2, 128.4, 128.5, 127.5, 127.4, 123.0, 116.9, 112.4, 112.3, 111.7, 100.8, 56.3, 42.5, 42.5.

*N′-(4-(benzyloxy)benzylidene)-4-(5-methoxy-1H-benzo[d]imidazol-2-yl)benzohydrazide* (**2i**): Yield: (85%); m.p.: 310–311 °C. IR ν cm^−1^ (KBr disk): 3440 cm^−1^ (NH stretch), 3164 cm^−1^ (Ar.C-H stretch), 1659 cm^−1^ (C = O), 1582 cm^−1^ (C = N), 1552 cm^−1^ (C-N), 1191 (C-O); ^1^HNMR (500 MHz, DMSO-*d*_6_): 11.8(s, 1H, NH), 8.7(s, 1H, NH), 8.3(s, 1H, N = CH), 8.2(d, *J* = 7.0 Hz, 2H, Aromatic H), 7.8(d, *J* = 6.4 Hz, 2H, Aromatic H), 7.7(m, 2H, Aromatic H), 7.2(d, *J* = 5.9 Hz, 3H, Aromatic H), 6.9(d, *J* = 6.6 Hz, 1H, Aromatic H), 6.8(s, 2H, Aromatic), 3.7(s, 3H, OCH_3_), 3.0(m, 6H, NCH3),. ^13^CNMR (125 MHz, DMSO-*d*_6_): δ 164.1, 161 8, 156.8, 153.2, 147.4, 140.2, 138.4, 137.1, 134.4, 132.9, 130.1, 130.1, 130.2, 130.2, 129.1, 129.1, 127.9, 127.8, 127.7, 127.4, 127.1, 127.0, 116.9, 113.9, 113.8, 109.8, 100.5, 72.1, 56.6.

*4-(5-methoxy-1H-benzo[d]imidazol-2-yl)-N′-(naphthalen-1-ylmethylene)benzohydrazide* (**2j**): Yield (80%); m.p.: 303–304 °C. IR ν cm^−1^ (KBr disk): 3392 cm^−1^ (NH stretch), 3179 cm^−1^ (Ar.C-H stretch), 1645 cm^−1^ (C = O), 1576 cm^−1^ (C = N), 1552 cm^−1^ (C-N), 1173 (C-O); ^1^HNMR (500 MHz, DMSO-*d*_6_): 12.1(s, 1H, NH), 9.0(s, 1H, NH), 8.8(s, 1H, N = CH), 8.2(s, 1H, Aromatic H), 8.0(m, 6H, Aromatic H), 7.8(d, *J* = 7.2 Hz, 2H, Aromatic H), 7.6(m, 3H, Aromatic H), 7.5(d, *J* = 6.8 Hz, 1H, Aromatic H), 7.3(s, 1H, Aromatic H), 2.9(s, 3H, OCH_3_). ^13^CNMR (125 MHz, DMSO-*d*_6_): δ 164.5, 156.8, 152.9, 146.9, 140.0, 138.6, 136.5, 134.3, 134.0, 133.1, 129.9, 129.8, 129.8, 128.4, 128.1, 128.1, 127.8, 127.7, 127.1, 127.0, 126.4, 126.3, 117.3, 111.8, 100.6, 56.8.

*N′-(3-bromo-2,4-dimethoxybenzylidene)-4-(5-methoxy-1H-benzo[d]imidazol-2-yl)benzohydrazide* (**2k**): Yield (75%); m.p.: 310–311 °C. IR ν cm^−1^ (KBr disk): 3405 cm^−1^ (NH stretch), 3160 cm^−1^ (Ar.C-H stretch), 1655 cm^−1^ (C = O), 1573 cm^−1^ (C = N), 1552 cm^−1^ (C-N), 1182 (C-O), 742 (C-Br); ^1^HNMR (500 MHz, DMSO-*d*_6_): 12.0(s, 1H, NH), 8.9(s, 1H, NH), 8.7(s, 1H, N = CH), 8.2(d, *J* = 6.7 Hz, 2H, Aromatic H), 8.1(d, *J* = 6.2 Hz, 2H, Aromatic H), 7.6(s, 1H, Aromatic H), 7.5(d, *J* = 6.2 Hz, 1H, Aromatic H), 7.4(s, 2H, Aromatic H), 7.3(d, *J* = 6.1 Hz, 1H, Aromatic H), 3.4(s, 9H, OCH_3_). ^13^CNMR (125 MHz, DMSO-*d*_6_): δ 164.8, 157.1, 155.0, 153.2, 149.6, 146.5, 139.8, 138.6, 134.4, 133.0, 130.1, 130.0, 127.9, 127.8, 119.3, 117.3, 116.7, 116.0, 114.9, 110.9, 101.5, 56.8, 55.7, 55.2.

*N′-(3-hydroxy-2,4-dimethoxybenzylidene)-4-(5-methoxy-1H-benzo[d]imidazol-2-yl) benzohydrazide* (**2l**): Yield (78%); m.p.: 317–318 °C. IR ν cm^−1^ (KBr disk): 3512 cm^−1^ (OH stretch), 3365 cm^−1^ (NH stretch), 3170 cm^−1^ (Ar.C-H stretch), 1658 cm^−1^ (C = O), 1574 cm^−1^ (C = N), 1552 cm^−1^ (C-N), 1168 (C-O); ^1^HNMR (500 MHz, DMSO-*d*_6_): 12.2(s, 1H, NH), 8.4(s, 1H, N = CH), 8.2(d, 6.2 Hz, 2H, Aromatic H), 7.7(m, 2H, Aromatic H), 7.6(d, *J* = 6 Hz, 2H, Aromatic H), 7.2(d, *J* = 6.1 Hz, 1H, Aromatic H), 6.7(d, *J* = 6.1 Hz, 1H, Aromatic H), 6.6(s, 1H, Aromatic H), 3.0(s, 9H, OCH_3_). ^13^CNMR (125 MHz, DMSO-*d*_6_): δ 164.6, 156.5, 152.9, 149.5, 149.3, 147.0, 140.1, 139.9, 138.3, 134.6, 132.7, 129.9, 129.9, 128.5, 127.5, 127.4, 116.8, 110.9, 104.8, 104.7, 101.5, 56.1, 56.1, 55.9.

*N′-(2,4-dimethoxybenzylidene)-4-(5-methoxy-1H-benzo[d]imidazol-2-yl)benzohydrazide* (**2m**): Yield (81%); m.p.: 307–308 °C. IR ν cm^−1^ (KBr disk): 3438 cm^−1^ (NH stretch), 3164 cm^−1^ (Ar.C-H stretch), 1656 cm^−1^ (C = O), 1576 cm^−1^ (C = N), 1554 cm^−1^ (C-N), 1170 (C-O); ^1^HNMR (500 MHz, DMSO-*d*_6_): 12.0(s,1H, NH), 11.6(s, 1H, NH), 8.5(s, 1H, N = CH), 8.3(d, *J* = 6.3 Hz, 2H, Aromatic H), 8.1(m, 3H, Aromatic H), 7.7(d, *J* = 6 Hz, 1H, Aromatic H), 7.4(d, *J* = 6.8 Hz, 1H, Aromatic H), 6.77(d, *J* = 6.6 Hz, 2H, Aromatic H), 6.70(d, *J* = 6.3 Hz, 1H, Aromatic H), 2.98(s, 9H, OCH_3_). ^13^CNMR (125 MHz, DMSO-*d*_6_): δ 164.1, 163.5, 159.9, 156.8, 153.0, 146.5, 139.9, 138.2, 134.7, 133.4, 132.9, 130.1, 130.1, 128.1, 128.0, 116.8, 112.0, 109.6, 105.9, 102.3, 100.7, 56.1, 55.9, 55.9.

### 3.3. Docking Studies

The intermolecular binding modes between benzimidazole thiosemicarbazide derivatives 1 (**a**–**j**) and benzimidazole Schiff bases 2 (**a**–**m**), and the active residues of acetylcholinesterase have been examined using Autodock package [26]. The acetylcholinesterase and its original docked donepezil geometries were download from the RCSB data bank web site (PDB code 4EY7) [27]. The active site is identified based on co-crystallized receptor–ligand complex structure of acetylcholinesterase. The re-docking of the original ligand donepezil into the active site is well reproduced with a RMSD value of 0.63 Å. Molecular geometries of benzimidazole-based analog derivatives were minimized at Merck molecular force field 94 (MMFF94) level44, and saved as PDB files. A docking study was performed using Lamarckian genetic algorithm, with 500 as the total number of run for the binding sites for the original ligands of the synthesized derivatives. In each respective run, a population of 150 individuals with 27,000 generations and 250,000 energy evaluations were employed. The operator weights for crossover, mutation, and elitism were set to 0.8, 0.02, and 1, respectively. The docking calculations have been carried out using an Intel (R) Core (TM) i5-3770 CPU @ 3.40 G Hz workstation.

## 4. Conclusions

In this study, ten analogs of benzimidazole thiosemicarbazide **1** (**a**–**j**) and 13 analogs of benzimidazole Schiff bases **2** (**a**–**m**) were synthesized and characterized through ^13^C NMR, ^1^H, and HR-EIMS. All the synthesized analogs of the two series were examined for their acetylcholinesterase and butycholinesterase inhibitory potentials which exhibited varying degrees of biological activities. Amongst the synthesized analogs of series **1, a**–**j**), **1b**, **1c**, and **1g** having IC_50_ values 1.30 ± 0.10, 0.60 ± 0.50, and 2.40 ± 0.10 µM, respectively, exhibited excellent acetylcholinesterase inhibition potential as compared to their inhibition potential to donepezil, and these analogs also showed excellent butycholinesterase inhibition with IC_50_ values of 2.40 ± 0.10, 1.50 ± 0.10 µM, and 2.40 ± 0.10 µM as compared to their inhibition potential to donepezil. While other analogs of the series showed good to moderate activities (Table 1), the rest of the compounds of this series also showed moderate to weak inhibition. The analogs of the second series **2** (**a**–**m**) compounds **2c**, **2e**, and **2h** having IC_50_ values of 1.50 ± 0.10, 0.60 ± 0.05, and 0.90 ± 0.05 µM, respectively, showed excellent acetylcholinesterase inhibitory potentials and also the same analogs **2c**, **2e**, and **2h** with IC_50_ values of 4.10 ± 0.10, 2.20 ± 0.10, and 2.20 ± 0.30 µM showed that this series displayed excellent butycholinesterase inhibition when compared to the standard drug donepezil (Table 2). The other analogs of this series showed good to moderate activity. SAR studies of these analogs was established through molecular simulation studies.

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
