# Peer review of "Synthesis of Benzimidazole–Based Analogs as Anti Alzheimer’s Disease Compounds and Their Molecular Docking Studies"

_molecules, 2020, doi:10.3390/molecules25204828_

Round 1
Reviewer 1 Report
AChE and BChE are considered as targets for the development of therapeutics of Alzheimer disease. A lots of highly potent inhibitors have been reported. This paper presents a series of compounds which have the AChE and BChE inhibitory activities. The topic is of interest but the contents does not qualify for acceptance based on following points:
1. The syntheses and the structures of compounds showed in this paper are not complicated but only 10 compounds have been studied. The number of compounds is too few and the structure lacks enough novelty.
2. Compare to the reported inhibitors and the positive control, the activities of compounds synthesized in this paper are two weak, the result is meaningless for reference. Based on the few number and poor activity, the discussion of structure-activity relationship makes nonsense.
3. The activities of synthesized compounds are only evaluated in AChE and BChE inhibition, which are not enough to characterize their anti-AD activities. More detailed investigation such as in related cell lines is suggested.
4. others: 1) Line 118, 1c and 1g have very similar AChE IC50 and BchE IC50 values. This data does not seem to support the conclusion that enzyme inhibition activity mainly depends on the substitution pattern of chloro group on the phenyl ring. In order to further study the structure-activity relationship, it is suggested to test the data of compounds substituted by single chloro group in ortho-, meta-, or para- substituted compounds on the phenyl ring. 2) Line 158, analog 2f does not seem to show enough selectivity under the condition that AchE IC50 and BchE IC50 are of the same order of magnitude.
Author Response
Respected Editor,
Thank you for giving us the opportunity for revision and considering our manuscript Manuscript ID: molecules-918474; Synthesis of benzimidazole based analogs as Alzheimer’s disease drug and their molecular docking studies, for possible publication in ''Molecules''. We are also thankful to the reviewers whose suggestion will definitely improve the quality of the manuscript. A detailed point by point response is here for your consideration. The changes made are highlighted in the text.
Reviewer 1
Comments and Suggestions for Authors
AChE and BChE are considered as targets for the development of therapeutics of Alzheimer disease. A lots of highly potent inhibitors have been reported. This paper presents a series of compounds which have the AChE and BChE inhibitory activities. The topic is of interest but the contents does not qualify for acceptance based on following points:
- The syntheses and the structures of compounds showed in this paper are not complicated but only 10 compounds have been studied. The number of compounds is too few and the structure lacks enough novelty.
Response to the reviewer: The total number of compounds reported comprising the two series are 10+13=23. The structural analogs are novel and most of them show powerful activities while some also show moderate activities.
- Compare to the reported inhibitors and the positive control, the activities of compounds synthesized in this paper are two weak, the result is meaningless for reference. Based on the few number and poor activity, the discussion of structure-activity relationship makes nonsense.
Response to the reviewer: Some analogs of 1st series, for example (1b, 1c and 1g) showed potent BChE inhibition when compared to donepezil having an IC50 value of 2.40 ± 0.10 µM, 1.50 ± 0.10 µM and 2.40 ± 0.10 µM. The rest of the compounds of this series also showed moderate to weak inhibition. While amongst the second series of analogs 2 (a-m), compounds 2c, 2e, 2h having IC50 value of 1.50 ± 0.10 µM, 0.60 ± 0.05, and 0.90± 0.05 µM respectively showed potent BChE inhibition when compared to Donepezil. Some compounds, for example 2c, 2e, 2h with IC50 value of 4.10±0.10 µM, 2.20±0.10 µM, 2.20±0.30 µM showed potent BChE inhibition when compared to donepezil.
- The activities of synthesized compounds are only evaluated in AChE and BChE inhibition, which are not enough to characterize their anti-AD activities. More detailed investigation such as in related cell lines is suggested.
Response to the reviewer: respected reviewer due to current situation we do not have facilities for desired experiment
- Others: 1) Line 118, 1cand 1ghave very similar AChE IC50 and BchE IC50 values. This data does not seem to support the conclusion that enzyme inhibition activity mainly depends on the substitution pattern of chloro group on the phenyl ring. In order to further study the structure-activity relationship, it is suggested to test the data of compounds substituted by single chloro group in ortho-, meta-, or para- substituted compounds on the phenyl ring. 2) Line 158, analog 2f does not seem to show enough selectivity under the condition that AchE IC50 and BchE IC50 are of the same order of magnitude.
Response to the reviewer: The compounds, 1c and 1g have different AChE and BChE activities. The AChE values are 0.60 ± 0.05 and 0.80 ± 0.05, while their BChe values are 1.50 ± 0.10 and 2.40 ± 0.10, which signifies the importance chloro group orientation on ph ring.
Reviewer 2 Report
In the manuscript titled “Synthesis of benzimidazole based analogs as Alzheimer’s disease drug and their molecular docking studies” the authors have come up with a different analog that could potentially be considered for further studies with AD. Although the work looks interesting, the analogs found are less potent when compared to the available drug. So I recommend this manuscript for minor revision before considering it for publication.
- It is important to add the units to the IC50 values wherever it occurs.
- An overlay of the docked poses of all the best compounds along with known donepezil should be shown in one figure (can be in the top of Figure 2) to better understand the difference in the binding orientation and the significance of various substituents at different positions.
- In both Figure 2 and 3, the stick representation is not clear.
Author Response
The manuscript titled “Synthesis of benzimidazole based analogs as Alzheimer’s disease drug and their molecular docking studies” the authors have come up with a different analog that could potentially be considered for further studies with AD. Although the work looks interesting, the analogs found are less potent when compared to the available drug. So I recommend this manuscript for minor revision before considering it for publication.
- It is important to add the units to the IC50values wherever it occurs.
Response to the reviewer: The IC50 units (µM) are incorporated throughout the manuscript.
- An overlay of the docked poses of all the best compounds along with known donepezil should be shown in one figure (can be in the top of Figure 2) to better understand the difference in the binding orientation and the significance of various substituents at different positions.
Response to the reviewer: The docking of the best actives compounds for both benzimidazole thiosemicarbazides (1c, 1b and 1g), benzimidazole Schiff bases (2c, 2h and 2e) and Donepezil into the active binding site acetylcholinesterase were added in a new Figure (Figure 1). Thank you for your remark.
- In both Figure 2 and 3, the stick representation is not clear.
Response to the reviewer: The stick representation of the docked compounds in Figures 2 and 3 are replaced by stick and ball representation, which are clearer. Thank you.
Reviewer 3 Report
The paper describes the synthesis of benzimidazole based analogs as Alzheimer’s disease drugs and their molecular docking studies. The products described are new, however their characterisation is inadequate as it lacks melting points and IR spectroscopical data.
Corrections:
- Spelling mistake in the title. It should be "drugs" and not "drug".
- Editing of the english language is required.
- No supporting information file provided. Please provide the NMR spectra of all new compounds.
- In scheme 1 the appearance of structure IV is not consistent with that of structure III or 1. Please make the structures consistent having the benzimidazole unit on the left.
Author Response
The paper describes the synthesis of benzimidazole based analogs as Alzheimer’s disease drugs and their molecular docking studies. The products described are new, however their characterisation is inadequate as it lacks melting points and IR spectroscopical data.
Response to the reviewer: We have provided IR and Melting point data
Corrections:
- Spelling mistake in the title. It should be "drugs" and not "drug".
Response to the reviewer: Corrected in the title and highlighted.
- Editing of the English language is required.
Response to the reviewer: The paper is revised and English is edited where required.
- No supporting information file provided. Please provide the NMR spectra of all new compounds.
Response to the reviewer: The NMR spectra are provided as supporting information.
- In scheme 1 the appearance of structure IV is not consistent with that of structure III or 1. Please make the structures consistent having the benzimidazole unit on the left.
Response to the reviewer: In scheme 1 the appearance of structure IV is now consistent with that of structure III or 1.
Round 2
Reviewer 1 Report
The revised ms added molecular modeling result to explain binding mode, the novelty is still not high enough. The poor enzymatic activity can not support the importance of compound, or the compound to be potential candidate of anti-Alzheimer desease. Besides, the following deficencies exist:
- The abstract is lengthy,it should be more concise and informative.
- "Alzheimer's desease drugs" is inappropriate to be used in the title since only weak and enzymes' inhibitors have been provided in the manuscript, it's far from "drug".
Author Response
- The abstract is lengthy, it should be more concise and informative.
Response: We have revised as suggested by respected reviewer
- "Alzheimer's desease drugs" is inappropriate to be used in the title since only weak and enzymes' inhibitors have been provided in the manuscript, it's far from "drug"
Response: We have revised as suggested by respected reviewer
Reviewer 3 Report
The authors have included a supporting information file but it is very basic. The SI file needs to have a cover page and each spectrum should have a title, the peak values (pick picking missing) and extra zoomed in pictures of the regions that are not clear need to be added.
It is noted that melting points and IR data have been added.
After this is done the paper can be accepted.
Author Response
The authors have included a supporting information file, but it is very basic. The SI file needs to have a cover page and each spectrum should have a title, the peak values (pick picking missing) and extra zoomed in pictures of the regions that are not clear need to be added.
Response: We have provided with covering page and provided also extra zoom in pictures wherever needed